# Contactless AC/DC Wide-Bandwidth Current Sensor Based on Composite Measurement Principle

**DOI:** 10.3390/s22207979

**Published:** 2022-10-19

**Authors:** Xiangyu Tan, Wenyun Li, Xiaowei Xu, Gang Ao, Fangrong Zhou, Jingjing Zhao, Qinghua Tan, Wenbin Zhang

**Affiliations:** 1Electric Power Research Institute, Yunnan Power Grid Co., Ltd., Kunming 650217, China; 2China Southern Power Grid, Yunnan Power Grid Co., Ltd., Kunming 650217, China; 3Yunnan Power Grid Co., Ltd., Kunming Power Supply Bureau, Kunming 650001, China; 4College of Science, Kunming University of Science and Technology, Kunming 650504, China; 5College of Mechanical and Electrical Engineering, Kunming University of Science and Technology, Kunming 650504, China

**Keywords:** contactless, current sensor, composite sensor, *Rogowski coil*, tunneling reluctance

## Abstract

With the accelerated construction of the smart grid, new energy sources such as photovoltaic and wind power are connected to the grid. In addition to power frequency, the current signal of power grid also includes several DC signals, as well as medium-high and high-frequency transient signals. Traditional current sensors for power grids are bulky, have a narrow measurement range, and cannot measure both AC and DC at the same time. Therefore, this paper designs a non-intrusive, AC-DC wide-bandwidth current sensor based on the composite measurement principle. The proposed composite current detection scheme combines two different isolation detection technologies, namely tunneling reluctance and the *Rogowski coil*. These two current sensing techniques are complementary (tunneling magnetoresistive sensors have good low-frequency characteristics and *Rogowski coils* have good high-frequency characteristics, allowing for a wide detection bandwidth). Through theoretical and simulation analysis, the feasibility of the composite measurement scheme was verified. The prototype of composite current sensor was developed. The DC and AC transmission characteristics of the sensor prototype were measured, and the sensitivity and linearity were 11.96 mV/A, 1.14%, respectively. Finally, the sweep current method and pulse current method experiments prove that the designed composite current sensor can realize the current measurement from DC to 17 MHz.

## 1. Introduction

The smart and clean grid is the inevitable trend of future power grid development [1,2]. In the application of the smart grid, it is necessary to use high-frequency semiconductor devices and converters to achieve power conversion and energy distribution [3,4]. Technology based on capacitive coupling can be used for the non-contact measurement of voltage [5,6,7,8], while non-intrusive current sensing techniques are still evolving. Compared with the traditional power grid, the current signal in the smart grid includes several DC, medium-, and high-frequency transient signals, as well as high-frequency transient signals in addition to power frequency and medium- and low-frequency signals. The current frequency band changes greatly from DC to tens of MHz [9,10]. The traditional measurement technology based on electromagnetic transformer is not suitable for measuring the current in a high-frequency converter. The shunt can realize the current measurement from DC to the frequency band, but it heats up severely under the action of high currents, leading to measurement errors. If the measurement accuracy is to be met, then the size and cost of the shunt increases. Moreover, the shunt must be connected in series in the circuit under test, which is a major safety hazard [11]. Therefore, the non-contact current sensing measurement technology which can realize AC/DC wide bandwidth has become a new direction of detection and monitoring in the modern power grid. Sensors and *Rogowski coils* [12,13] based on magnetic induction are the most popular technologies for isolating current detection. The Roche coil is a magnetic coupling sensor which uses Faraday’s law of electromagnetic induction to detect the current in the conductor in an isolated way. The *Rogowski coil* provides an output voltage proportional to the time derivative of the input current, integrating the output signal with active or passive circuits to derive the actual current information [14,15]. The *Rogowski coil* generally has no magnetic core and has inherent wide-bandwidth response characteristics [16]. However, the *Rogowski coil* cannot respond to constant and DC fields, which makes it unsuitable for DC measurement applications. Compared with the *Rogowski coil*, the current sensor based on the magnetoresistive element can realize the current measurement from DC to higher frequency. However, most magnetoresistive components are affected by the characteristics of the material itself, and the measurement bandwidth is limited. The maximum measurement frequency is usually only several hundred kHz [17,18]. In order to realize the current measurement from DC to high frequency, some scholars have proposed a composite measurement method for measuring the current in the power electronic system [19,20]. However, in the power system, this current measurement technology based on composite principle has not been reported in relevant research and application.

Based on this, aiming at the non-contact measurement of the AC/DC wide-bandwidth current in the power grid, a non-contact AC/DC wide-bandwidth composite current sensor was designed by combining tunneling reluctance (TMR) and the *Rogowski coil*. The low-frequency current sensing element was measured by the high-bandwidth TMR sensor, and the high-frequency current sensing element was measured by the open *Rogowski coil* designed on the PCB, which can be easily closed and opened, and can achieve the purpose of spread spectrum. Finally, the feasibility of the design was verified by circuit simulation and experimental test.

The non-contact AC-DC broadband composite current sensor designed in this paper has the advantages of low power consumption, easy application, isolation from the measurement circuit, low insulation design requirements, and simultaneous AC-DC measurement, which can be used for commutation current measurement in fast-switching power electronics in smart grids to monitor the operation of the grid in real-time.

## 2. Principle of Current Measurement

### 2.1. Principle of Current Sensor Based on Magnetoresistive Element

According to the Biot–Savart law, the magnetic field produced by the current carrying wire at a point in space *p* is:(1)$$\vec{B}=\int d\vec{B}=\int \frac{\mu_{0}I}{4\pi }\frac{d\vec{l}\times \vec{r}}{r^{3}}$$

As shown in Figure 1, the current passing through the interior of the current-carrying long straight conductor CD is I, and the distance between a point *p* in the space and conductor CD is *r*_0_. It can be concluded that the magnetic field generated by conductor CD at point *p* is:(2)$$B=\frac{\mu_{0}I}{4\pi }\int_{\theta_{1}}^{\theta_{2}}\sin\theta d\theta =\frac{\mu_{0}I}{4\pi }(\cos\theta_{1}-\cos\theta_{2})$$

Therefore, when the length of wire A is constant and the position of wire and *p* point is relatively fixed, the magnetic field B generated by wire A at *p* point is proportional to the current value inside the wire. Figure 2 is the measurement principle diagram of the TMR current sensor. When the current *I* flows through the conductor, according to the right-handed helix rule, a spiral magnetic field will be generated around the conductor. The magnetic field is proportional to the current inside the conductor. The TMR chip is fixed around the conductor, and the direction of the magnetic field generated by the conductor is parallel to the direction of the sensitive axis of the TMR chip. The TMR sensor will change its resistance according to the magnetic field, as shown in Figure 3a. The internal structure of the TMR chip is a Wheatstone bridge, and the change of the magnetic field is converted to the corresponding output voltage signal, as shown in Figure 3b [21]. There is parasitic capacitance in TMR element, and its internal equivalent circuit model is shown in Figure 4a, where $R_{MTJ}$ is the resistance of each reluctance in the bridge, and $C_{P}$ is the parasitic capacitance on the reluctance. From the output side, this model can be further simplified as the RC circuit shown in Figure 4b. In the figure, the resistance $R_{MTJ}$ and a capacitor $C_{P}$ form a low-pass filter, so the system transfer function of the TMR sensor can be expressed as:(3)$$H_{l}=\frac{1}{1+2\pi R_{MTJ}C_{P}}$$

It can be seen from Figure 4b and Formula (3) that the output voltage signal of each TMR sensor is proportional to the measured current below the cutoff frequency. The cut-off frequency is defined by the resistance and parasitic capacitance of the TMR element. Therefore, adding parallel resistors to the differential output of the TMR sensor can increase the bandwidth, but the disadvantage of this method is that it reduces the sensitivity of the sensor.

### 2.2. Measurement Principle of the Planar PCB Roche Coil Current Sensor

The planar PCB *Rogowski coil* has the advantages of measurement, high precision, good linearity, and high-frequency performance [22]. The planar PCB type *Rogowski coil* structure, as shown in Figure 5, i(t) is the current to be measured of the current overload current conductor, *a* and *b* are the inner diameter and outer passage of the coil, *r* is the distance from the center of the current carrying conductor on the PCB, *h* is the height of the PCB board, and *N* is the number of turns of the coil. Therefore, the total flux through the PCB conductive trace is:(4)$$\Phi =\int_{a}^{b}N\mu_{0}\times \frac{i(t)}{2\pi r}\times dr\times h=\frac{N\mu_{0}i(t)h}{2\pi }\times \ln\frac{b}{a}$$
where $\mu_{0}=4\pi \times 10^{-7}H/m$ is the vacuum permeability.

According to the Biot–Savart law and Faraday’s law, when the measured current changes, the induced potential of the coil is:(5)$$e(t)=-\frac{Nu_{0}h}{2\pi }\ln\frac{b}{a}\frac{di(t)}{dt}=-M\frac{di(t)}{dt}$$
where *e*(*t*) is the induced electromotive force of the coil, and $M=(Nu_{0}h/2\pi )\ln(b/a)$ is the mutual inductance between the coil and the current in the measured conductor. In addition, since the winding line of the *Rogowski coil* is spirally wound on the skeleton, there will be self-inductance. In the planar PCB *Rogowski coil* structure shown in Figure 5, self-inductance can be expressed as: $L_{0}=(\mu N^{2}h/2\pi )\ln(b/a)$. It can be seen that the self-inductance and mutual inductance of the *Rogowski coil* depend on the geometric shape, size parameters, and turns of the coil [23]. At the same time, it can be obtained from Equation (5) that the induced voltage *e*(*t*) has no bandwidth limitation theoretically, and even increases under high-frequency signals. The induced voltage *e*(*t*) of the *Rogowski coil* is proportional to the change rate of the measured signal over time.

Figure 6 is the *Rogowski coil* equivalent circuit based on the lumped parameter model (LPM). This model is effective before the first resonant frequency [24,25]. The equivalent circuit of the lumped parameter model (LPM) consists of the voltage source *e*(*t*), parasitic inductance $L_{0}$, parasitic resistance $R_{0}$, and parasitic capacitance $C_{0}$, and load resistance *R*_0_. $L_{0}$ and $C_{0}$ limit the *Rogowski coil* bandwidth. Due to the existence of parasitic elements, the *Rogowski coil* has a second-order transfer function, which is expressed as:(6)$$H_{h}(s)=\frac{Ms}{L_{o}C_{o}(s^{2}+\frac{1}{L_{o}C_{o}}(R_{o}C_{o}+\frac{L_{0}}{R_{d}})s+\frac{1}{L_{o}C_{o}}(\frac{R_{0}}{R_{d}}+1))}$$

Reduce Equation (6) to the expression of a standard second-order system as follows:(7)$$H_{h}(s)=\frac{MR_{d}s}{\frac{1}{\omega_{0}^{2}}s^{2}+\frac{2\zeta }{\omega_{0}}s+1}$$

Comparing the denominator with a standard second-order equation, the angular frequency for the RLC circuit is:(8)$$\omega_{0}=\frac{1}{\sqrt{L_{0}C_{0}}}\sqrt{\frac{R_{d}+R_{0}}{R_{d}}}$$

When $R_{d}$>>$R_{0}$, the angular frequency depends only upon self-inductance and equivalent capacitance. The damping factor for the RLC circuit is:(9)$$\xi =\frac{1}{2\sqrt{L_{0}C_{0}}}\sqrt{\frac{R_{d}}{R_{d}+R_{0}}}(\frac{L_{0}}{R_{d}}+C_{0}R_{0})$$

In order to obtain fast rise time without oscillation, damping matching must be carried out on the circuit.

### 2.3. Measurement Principle of the Composite Current Sensor

In this paper, the HOKA principle was used to realize the combination of sensing principle. The ideal HOKA principle was proposed by Karrer and Hofer. Two kinds of sensors with different performances are needed in the principle. One type of sensor must have the characteristics of low-pass filter, and the other type of sensor must provide output signals proportional to the derivative of the measured current. It can be seen from the previous analysis that the transmission characteristics of the TMR current sensor system are low-pass characteristics, and it is suitable for low-frequency current measurement from DC to a certain frequency. The system transfer characteristic of the *Rogowski coil* is differential (proportional to the derivative of measured current) and is suitable for high-frequency current measurement. If the advantages of both are combined, DC to high-frequency wide bandwidth current measurement can be realized. Figure 7 shows the frame diagram of the composite inductor measurement scheme used in this paper.

The low-frequency measurement part of composite measurement is composed of current sensors based on the TMR chip. The ideal output signal of the TMR component is proportional to the measured current. However, the first-order inertia link formed by the resistance and parasitic capacitance of the reluctance itself limits the frequency band of the reluctance. Therefore, we can model the TMR current sensor as a low-pass filter, and then output the low-frequency measurement signal of the composite sensing system through an amplifier. The transfer function is:(10)$$H_{l}=K_{l}(\frac{1}{1+sT_{h}})(\frac{K_{h}}{K_{l}})$$
where $K_{l}$ is the sensitivity of the TMR sensor, $T_{h}$ is the time constant of the low-pass filter in the low-frequency measurement part, and $\frac{K_{h}}{K_{l}}$ is the amplification coefficient of the low-frequency measurement part.

The high-frequency measurement part of composite measurement is composed of a high-frequency current sensor based on the *Rogowski coil*. Before the first resonant frequency of the *Rogowski coil*, the output signal and the measured current signal form a differential relationship. After integration by a first-order low-pass filter, within the integration range, the output signal can be output in proportion to the measured current signal, and its transfer function is:(11)$$H_{h}(s)=(\frac{Ms}{T_{2}s^{2}+T_{1}s+T_{0}})(\frac{1}{sT_{h}+1})(\frac{K_{h}T_{h}}{M})$$
where $T_{2}=L_{0}C_{0}$, $T_{1}=R_{0}C_{0}+(L_{0}/R_{S})$, $T_{0}=1+(R_{0}/R_{S})$, $T_{h}$ is the time constant of the integrator for the high-frequency measurement part, and $K_{h}T_{h}/M$ is the magnification factor for the high-frequency measurement part.

The output signals of the low-frequency measurement part and the high-frequency measurement part are superimposed, and $T_{h}=T_{l}$ is defined to obtain the transfer function of the composite current sensor.
(12)$$H_{s}(s)=K_{h}\frac{(s^{2}T_{2}+sT_{1}+1)+sT_{h}}{\left(s^{2}T_{2}+sT_{1}+1)(sT_{h}+1\right)}$$

Finally, the amplitude-frequency response of the composite current sensor and the amplitude-frequency response of the low-frequency and high-frequency paths are shown in Figure 8.

It can be seen from the above figure that in the composite principle, the low-frequency sensor TMR detects the current from DC to a certain frequency. With the increase of frequency, the induced voltage generated by the high-frequency sensing element (*Rogowski coil*) due to the change of current will also increase. Through the adjustment of integration and amplification factor, the measurement signals of the two parts are superimposed under the same sensitivity to obtain the current measurement with wide-bandwidth characteristics. In this scheme, the low-frequency performance depends on the low-frequency sensing element TMR, and the high-frequency performance depends on the high-frequency response range of the high-frequency *Rogowski coil*, which is mainly related to the parasitic capacitance and parasitic inductance of the *Rogowski coil* [26].

## 3. Design and Implementation of Composite Current Sensor

### 3.1. Design of Loop Array TMR Current Sensor

The magnetic field generated by the Earth can be regarded as a magnetic field with uniform space and constant time. If only one magnetic field sensor is used, then the measured signal will produce error. If there are two uniaxial magnetic field sensors with the same sensitivity, their positions are relative, and the sensitivity axis is reversely parallel, then this problem can be solved. Two TMR sensors were symmetrically arranged around the middle hole (shown in Figure 9). The size of the Earth’s magnetic field detected by each sensor was the same, but the sensitivity axis was inversely parallel, and its polarity was opposite. By adding the two sensor signals, the error caused by the earth magnetic field can be eliminated.

In this paper, TMR2003 was used as the low-frequency magnetic field sensing element. According to the internal equivalent diagram of the magnetoresistive chip shown in Figure 4b, the equivalent impedance *Z*(*s*) formula in the s-domain circuit is:(13)$$Z(s)=\frac{R_{MTJ}}{R_{MTJ}C_{P}s+1}$$

The electrical parameters between the output pins of magnetoresistive chip TMR2003 were measured by the GWinstek LCR-8101 G impedance meter. The measured resistance of the magnetoresistive chip was $R_{MTJ}=57\;k\Omega$, and the impedance frequency response curve is shown in Figure 10. According to Formula 14, parasitic capacitance $C_{P}=24\;\mathrm{pF}$. Thus, the theoretical frequency response curve of impedance was obtained, as shown by the blue dotted line in Figure 10. By comparing the theoretical and measured impedance-frequency curves, it can be seen that the theoretical impedance was highly consistent with the measured impedance, so as to determine the electrical parameters of the internal structure equivalent circuit of TMR2003 and obtain the cutoff frequency of the chip system response of TMR2003:(14)$$f_{l}=\frac{1}{2\pi R_{TMR}C_{p}}=116\mathrm{kHz}$$

As the applied magnetic field changes parallel to the sensitive direction of the sensor, the Wheatstone full bridge provides differential voltage output. Hence, a differential amplifier circuit is needed at the back end to realize the single-ended output of the magnetoresistance signal. The conditioning circuit uses a high-speed differential operational amplifier AD8130 (270 MHz, 1090 V/us), so it does not limit the frequency range of low-frequency measurement of composite current sensors. Taking one of the circular arrays TMR low frequency current sensors as an example, the equivalent circuit diagram of the current sensor was built by Pspice simulation software. The rationality of the circuit was verified by the sweep analysis function in the software. The equivalent circuit built by Pspice software is shown in Figure 11.

In Figure 11, a current controlled voltage source and a low-pass filter were used to simulate the TMR sensor. The current controlled voltage source converted the measured current signal into a voltage signal, and the conversion ratio was related to the magnetic field sensitivity *k* of the TMR sensor and the distance *r* between the measured current and the TMR sensor. The magnetic field sensitivity of tmr2003 chip used in this paper was 60 mV/V/mT, and the power supply was 5 V. The distance between the center of the ring array current sensor and the TMR sensor was designed to be 0.02 m. Through calculation, the conversion ratio of current controlled voltage source was found to be 1:0.003. The specific calculation formula is as follows:(15)$$k^{\prime }=\frac{u_{0}k}{2\pi r}=\frac{\;4\pi \times 10^{-7}T\cdot m/A\times 300V/T}{2\pi \times 0.02}=3\mathrm{mV}/A$$

As shown in Figure 12, the frequency characteristic curve of a single TMR low-frequency current sensor was obtained by simulation. The simulation results show that the −3 dB bandwidth of the single TMR low-frequency current sensor was from DC to 116 kHz.

### 3.2. Design of the High-Frequency Rogowski coil Current Sensor

In order to facilitate the installation and disassembly without disconnecting the measuring line in practical use, the *Rogowski coil* is designed as an open structure. However, the open-ended PCB *Rogowski coil* is vulnerable to external magnetic field interference. It is necessary to consider whether the winding can be continuously and uniformly distributed at the opening. As shown in Figure 13, the symmetrical wiring method is used to make the two half-rings of the opening completely symmetrical, and the distance between the left and right turns of the winding at the opening is consistent with that of other windings, i.e., $d_{1}=d_{2}$. When the two half-rings are completely closed, the winding of the whole coil is evenly distributed, which enhances the ability to resist the interference of external magnetic field changes in the parallel direction. On the double-layer PCB board, two rows of holes in the inner ring and one row of holes in the outer ring are placed in each half of the coils. The small circle is the through hole, the red line is the interactive route at the top, and the blue line is the interactive route at the bottom. The path of clockwise winding is the same as that of counterclockwise winding, but the direction is opposite. They contain the same planar region, increasing the ability to resist the interference of externally varying magnetic fields in the vertical direction. A layer of copper wire is placed on the outermost ring of the winding to resist the influence of the external electric field on the coil.

In Figure 13, on the semi-ring PCB, the three rows of holes, from inside to outside, are A, B, and C. In addition, a transition hole D is added to each semi-ring, and a wiring in obtuse angle form is used at the vias for transition. The number of the hole starts counterclockwise, sorting from A1, B1, C1. The PCB coil with 20 turns in the upper half loop is taken as an example. The winding along the winding path is C1-A1-C3-A2 -... -C19-A10. After passing through D, the winding begins to turn back, and the winding path is D-B10-C20-B9-C18 -... -B1-C2. This winding method should keep the same direction of the wire winding skeleton when it is winding along and winding back. Otherwise, it will offset the measured induced electromotive force generated by the winding along and winding back. The winding wiring of the lower half-ring coil is the same as that of the upper half-ring coil, and E1 and E2 are connected to make the induction voltage of the upper and lower half-ring superimposed. F1 and F2 are the output terminals of the *Rogowski coil* signal.

The winding parameters of the coil designed in this paper are shown in Table 1.

There are two types of windings in the coil designed: AC and BC, whose total mutual inductance is the sum of the mutual inductances of the different types of turns. As with the total mutual inductance, the same is true for the self-inductance of the *Rogowski coil*. The two parameters are calculated as follows.
(16)$$M=N\times \frac{\mu \times h}{2\times \pi }\times (\ln\frac{b}{R_{A}}+\ln\frac{b}{R_{B}})$$
(17)$$L_{0}=N^{2}\times \frac{\mu \times h}{2\times \pi }\times (\ln\frac{b}{R_{A}}+\ln\frac{b}{R_{B}})$$

Internal resistance of the coil:(18)$$R_{0}=\rho \times \frac{l}{A}$$
where $\rho =1.68\times 10^{-8}\Omega /m$ is the resistivity of copper.

The lumped stray capacitance of a PCB Rogowski, which mainly includes the capacitance between the wires on the same side of the double panel, the capacitance distributed between the wires on both sides, and the shielding capacitance, can be obtained by complex numerical analysis or measurement. In this paper, the total stray capacitance of the coil was measured using experimental methods.

In the s-domain circuit, the equivalent impedance *Z*(*s*) of the *Rogowski coil* is derived as follows:(19)$$Z(s)=\frac{L_{0}s+R_{0}}{L_{0}C_{0}s^{2}+R_{0}C_{0}s+1}$$

The electrical parameters of the *Rogowski coil* were measured by the LCR3535 impedance meter from the HOKA company. Figure 14 shows the schematic diagram of the electrical parameter measurement experiment of the openable PCB *Rogowski coil*. The inductance of the *Rogowski coil* was 0.67 μH, and the resistance was 2.7 Ω. The lumped stray capacitance in the coil could not be directly measured. Usually, the first resonant frequency of the amplitude-frequency response of the impedance is used for theoretical derivation. The derivation formula is as follows:(20)$$C_{0}=\frac{1}{L_{0}\omega_{0}^{2}}\frac{R_{d}+R_{0}}{R_{d}}$$

The impedance of the *Rogowski coil* was measured by sweep frequency, and the measurement results are shown in Figure 15a. From the measured impedance sweep curve, it is known that the first resonance point of the coil’s impedance was at 89.8 MHz. According to Equation (20), the total set stray capacitance was 5.2 pF. According to Equation (19), the theoretical sweep frequency curve was obtained and compared with the measured impedance sweep frequency curve. The results are shown in Figure 15b.

It is known that the theoretical impedance agrees well with the measured impedance. In summary, the basic parameters of the coil set total parameter model were determined as shown in Table 2.

The output of the *Rogowski coil* is differential. In order to recover the current to be measured, it must be used with integrator. For high-frequency applications, passive integrators have more advantages than active integrators because they do not rely on passive components and gain bandwidth constraints. Therefore, this paper used a simple RC passive low-pass filter as an integrator with higher frequency. At the same time, a capacitor was changed to several two capacitors in parallel to reduce the series inductance of the capacitor and the refraction and reflection of the high-frequency signal. The specific high-frequency Roche line integral circuit is shown in Figure 16. When the TMR sensor is combined with *Rogowski coil*, it is necessary to select the cutoff frequency of the low-pass filter very carefully. In order to ensure the smooth superposition of low-frequency and high-frequency measurement signals in the measurement frequency band, the integral frequency of *Rogowski coil* needs to match the cutoff frequency of TMR sensor response. By defining $R_{1}=9.1k\Omega$, $C_{1}=75pF$, $C_{2}=75pF$, the cutoff frequency of RC integrator can be calculated as:(21)$$f_{h}=f_{l}=\frac{1}{2\pi R_{1}(C_{1}+C_{2})}=116kHz$$

The high-frequency signal above 116 kHz can be integrated.

### 3.3. Overall Design and Simulation of Composite Current Sensor

In order to integrate the composite current sensor and realize the wide bandwidth characteristics of the sensor, it is necessary to superimpose the low-frequency measurement signal and the high-frequency measurement signal through a composite circuit with scaling and addition functions. The composite current sensor composite circuit is shown in Figure 17.

In order to achieve the constant sensitivity of the sensor within the working frequency range, it is necessary to match the gain multiples of the low-frequency and high-frequency measurement signals to make them equal. Specific to the composite circuit, the ratios of R14/R52, R14/R53, R14/R59, R14/R63, and R14/R56 need to be adjusted. Combined with the parameters of the previously designed toroidal array TMR low-frequency current sensor, the openable PCB high-frequency Roche coil current sensor, and the corresponding back-end processing circuit, the specific settings of the electrical parameters in the composite circuit of the composite wide bandwidth current sensor are shown in Table 3.

Pspice circuit simulation software was used to build the overall circuit of the non-contact composite wide-bandwidth current sensor containing a ring array TMR low-frequency current sensor, an openable PCB-type high-frequency Roche coil current sensor, and a coupling circuit, as shown in Figure 18.

The overall frequency characteristics of the non-contact composite wide bandwidth current sensor were analyzed by the AC frequency sweep function of the simulation software, and the obtained amplitude-frequency response of the sensor is shown in the curves in Figure 19.

From the simulation results of the amplitude frequency response of the non-contact composite broadband current sensor, it can be seen that the amplitude frequency response of the designed non-contact composite broadband current sensor remained very stable from DC to 30 MHz, the gain remained at about −37.8 db, and the sensitivity was about 12.9 mv/a, which can realize the measurement of current signals with frequencies above DC to 30 MHz.

Figure 20 shows the hardware prototype of the designed non-contact composite broadband current sensor.

## 4. Experiment Results

### 4.1. DC Transmission Characteristics Test of Non-Contact Composite Wide-Bandwidth Current Sensor

The 1 A to 20 A DC current carrying wire passed through the current composite current sensor center. The response of the sensor to the change of the current amplitude under test was recorded. Figure 21 shows the DC response of the composite current sensor. The calculation sensitivity of the DC current response test of the composite current sensor through data processing was 11.96 mV/A, and the linearity was 1.14%.

### 4.2. AC Output Waveform Test

In order to obtain the performance of non-contact composite wide-band current sensor in terms of AC measurement, it is necessary to observe the consistency of input-output waveform in the AC measurement. The signal generator and power amplifier were used to generate 50 Hz, 500 Hz, 10 kHz, and 300 kHz sinusoidal currents of 500 mA (amplitude). The current wire was passed through the central circular hole of the non-contact composite broadband current sensor, and the reference current probe was clamped on the wire to measure the input current. The measured input signal and the output signal of the composite current sensor were connected to the oscilloscope together to observe their waveforms. Figure 22 shows the composite broadband current sensor in the measurement of the 50 Hz, 500 Hz, 10 kHz, and 300 kHz AC currents (with 500 mA amplitude), as well as the input and output waveform comparison diagram.

It can be seen from the waveform that the amplitude and phase of the input and output waveform maintained a good consistency under the conditions of the 50 Hz, 500 Hz, 10 kHz, and 300 kHz AC input current The results show that the non-contact composite broadband current sensor can better restore the measured signal when measuring the AC current signal.

### 4.3. Frequency Range Test

The amplitude-frequency response curve of the sensor obtained by the sweep current method can well reflect the sensing bandwidth and sensitivity of the current sensor. Figure 23 shows the experimental results of frequency sweep current method for the non-contact composite broadband current sensor. The green real line is the amplitude-frequency response curve of the low-frequency TMR current sensor, and its −3 dB bandwidth was DC-110 kHz. The blue real line is the amplitude-frequency response curve of the high-frequency *Rogowski coil* current sensor, which maintained good amplitude-frequency characteristics in the frequency band of 110 kHz–2 MHz. The red real line is the amplitude-frequency response curve of the composite current sensor, which maintained good amplitude-frequency characteristics in the frequency band of DC-2 MHz and had almost no attenuation. In the DC-2 MHz bandwidth range, the response amplitude of the non-contact composite broadband current sensor was about −38.5 dB.

In addition, the rectangular current waveform contains more harmonic components than other waveforms (such as triangular waveform), so it is very effective for current sensors. The pulse voltage source used in this paper, which has a rise time capable of reaching 20 ns, generates rectangular current pulse by an external inductanceless resistor. To display the low-frequency performance of the current sensor, a longer duration can be selected. The input current signal of the reference current probe (Tek TCP0030A) and the output signal of the non-contact composite broadband current sensor were transmitted by coaxial cables with characteristic impedance of 50Ω. The pulse current method test site was built as shown in Figure 24. For better observation, the input pulse current signal measured by the reference current probe and the output pulse voltage signal measured by the composite broadband current sensor were normalized to obtain the pulse signal response diagram of the non-contact composite broadband current sensor, as shown in Figure 25.

It can be seen from the figure that the response process of the reference current probe was highly matched with that of the composite current sensor. The measured signal of non-contact composite broadband current sensor showed high fidelity, but the measured signal oscillated. These oscillations were caused by the capacitance coupling between the composite current sensor and the pulse power supply, which were mainly concentrated near the rise and fall edges of the current pulse. The rise time of the reference current probe and the open sensor was measured to be 20.3 ns and 21.1 ns, respectively. According to the relationship between the rise time and the highest frequency of measurement, the high-frequency cutoff frequency of the measurement system was calculated to be >17 MHz. The calculation formula is as follows:(22)$$BW=\frac{0.35}{Tr}$$

## 5. Conclusions

In this paper, a non-contact AC/DC composite current sensor based on two complementary characteristics was designed using the advantages of good low-frequency characteristics of tunneling magnetoresistive sensor and good high-frequency characteristics of the *Rogowski coil*. The low-frequency measurement part of the composite current sensor was completed by the TMR2003 reluctance element, and the high-frequency measurement part was captured by the *Rogowski coil*. The designed composite current sensor combines these two sensing technologies to achieve broadband response from DC to several MHz. In this paper, the proposed sensing scheme is theoretically analyzed in detail, and the design of the proposed sensor is discussed in detail and verified by circuit simulation. Finally, through experiments, it was found that the designed composite current broadband current sensor had a sensitivity of 11.96 mV/A within the frequency range of DC-2MHz; under the pulse square wave impact, the time domain response waveform and the input waveform followed well, and the measured rise time was 21.1 ns, which could achieve the highest current measurement of 17 MHz.

However, the non-contact composite broadband current sensor designed in this paper is only a prototype. In other words, its noise, offset, dynamic range, power consumption, and area have not been optimized, and subsequent optimization needs to be carried out in combination with the specific application environment.

## Figures and Tables

**Figure 1 sensors-22-07979-f001:**
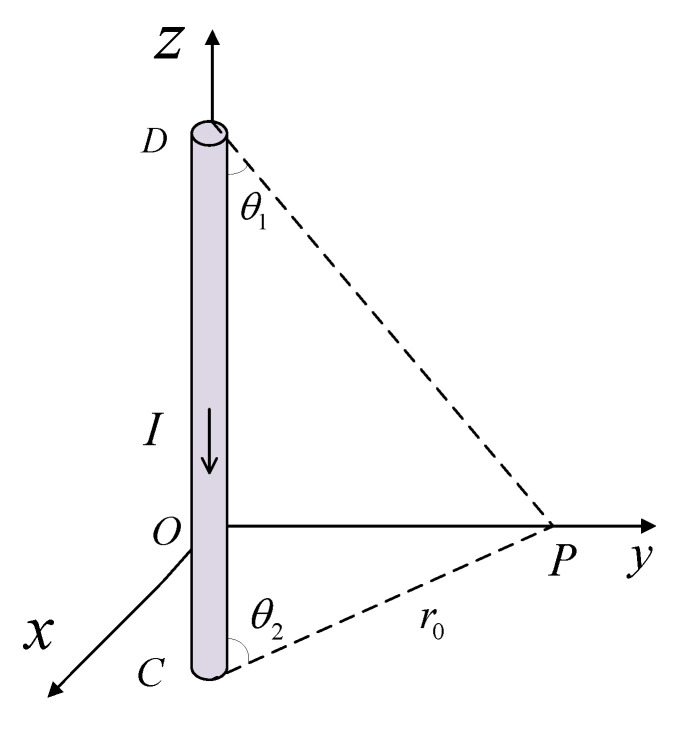
A schematic diagram of the magnetic field generated around a current-carrying wire.

**Figure 2 sensors-22-07979-f002:**
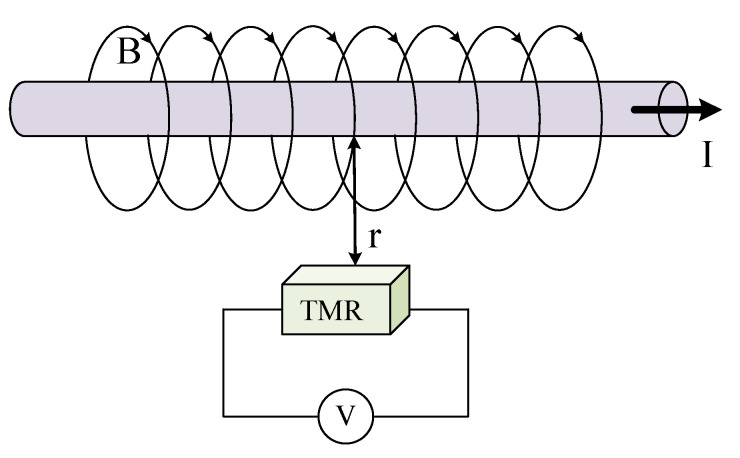
Measurement principle diagram of the TMR current sensor.

**Figure 3 sensors-22-07979-f003:**
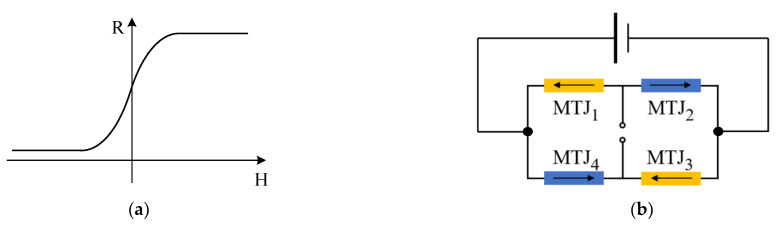
TMR sensing characteristics. (**a**) Resistance of magnetoresistance versus magnetic field, (**b**) Wheatstone bridge structure of the TMR sensor.

**Figure 4 sensors-22-07979-f004:**
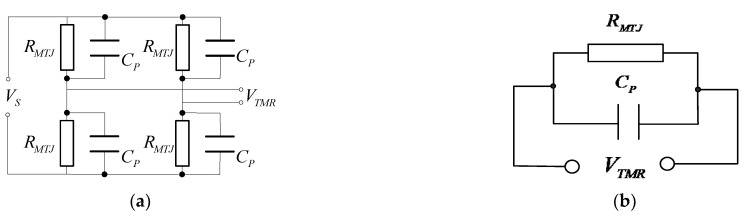
Structure of the TMR sensor. (**a**) Internal parasitic capacitance of the TMR sensor, (**b**) internal equivalent structure of the TMR sensor.

**Figure 5 sensors-22-07979-f005:**
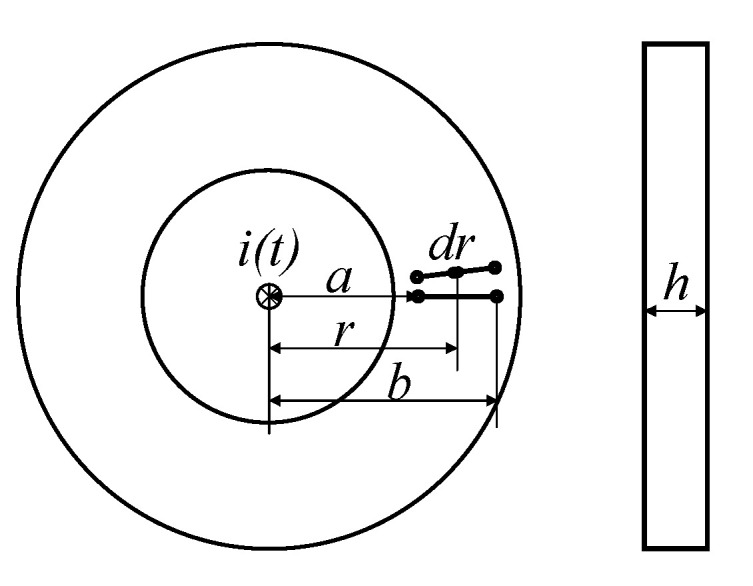
Structure of the planar PCB Roche coil.

**Figure 6 sensors-22-07979-f006:**
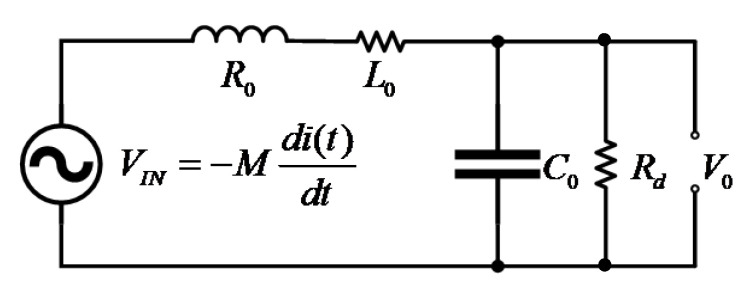
Lumped parameter model of the *Rogowski coil*.

**Figure 7 sensors-22-07979-f007:**
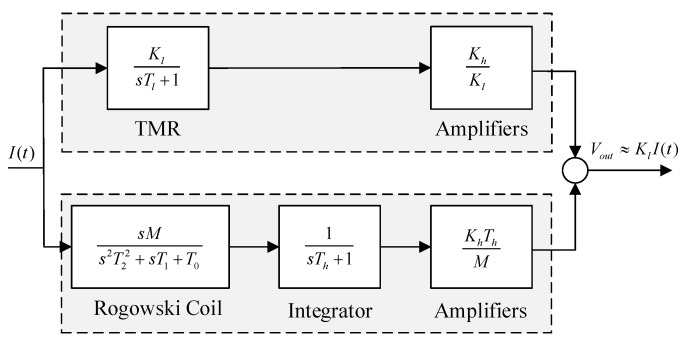
Principle of the composite current measurement.

**Figure 8 sensors-22-07979-f008:**
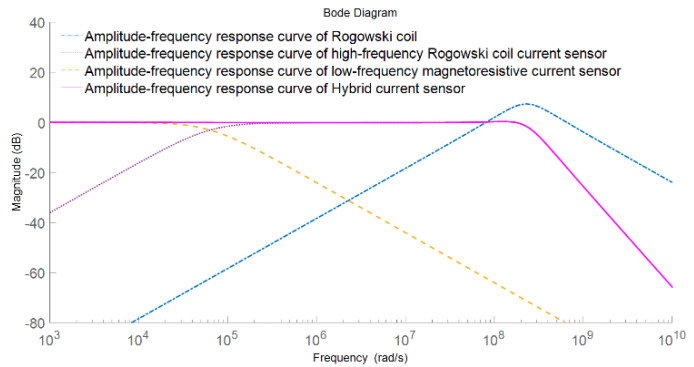
Amplitude-frequency response of the composite current measurement scheme.

**Figure 9 sensors-22-07979-f009:**
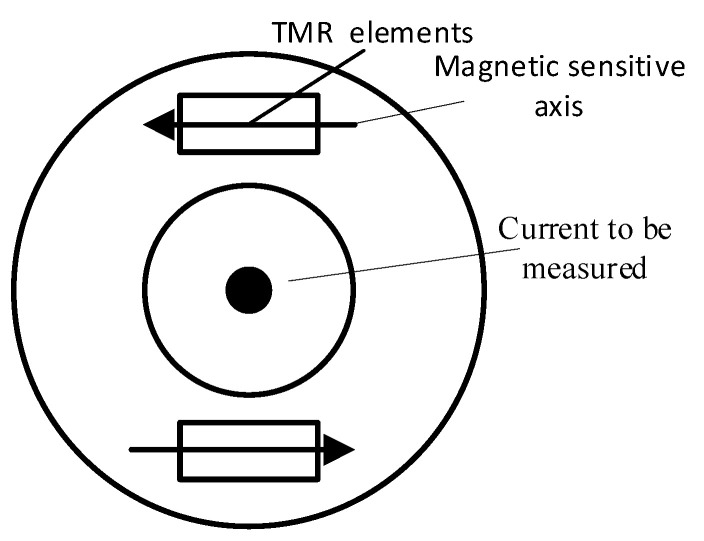
TMR sensor ring array layout diagram.

**Figure 10 sensors-22-07979-f010:**
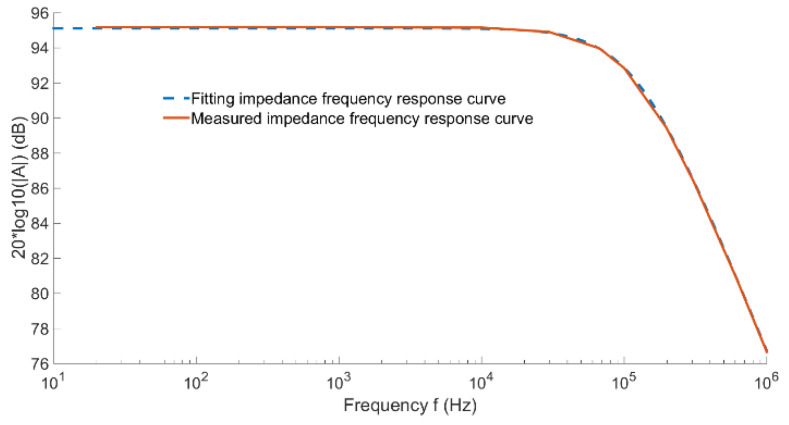
TMR2003 impedance frequency response curve.

**Figure 11 sensors-22-07979-f011:**
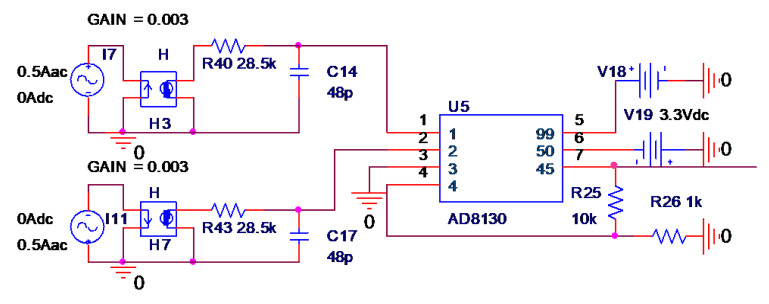
TMR low-frequency current sensor equivalent circuit simulation.

**Figure 12 sensors-22-07979-f012:**
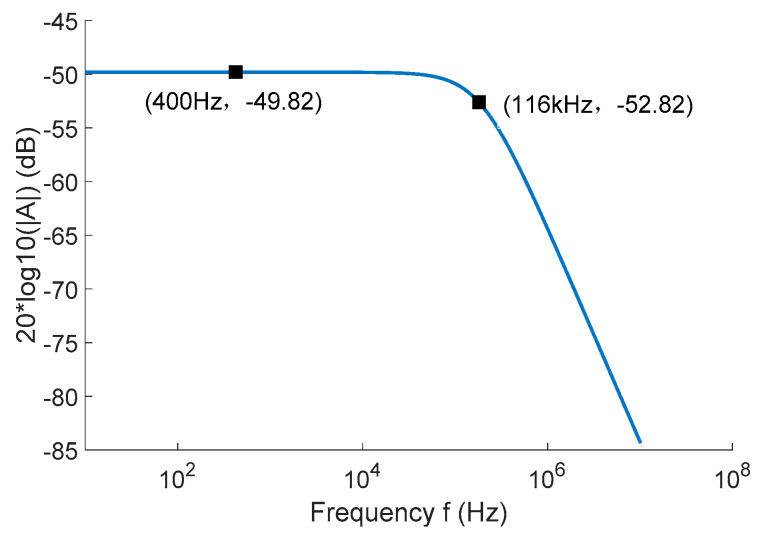
TMR low-frequency current sensor simulation frequency characteristic curve.

**Figure 13 sensors-22-07979-f013:**
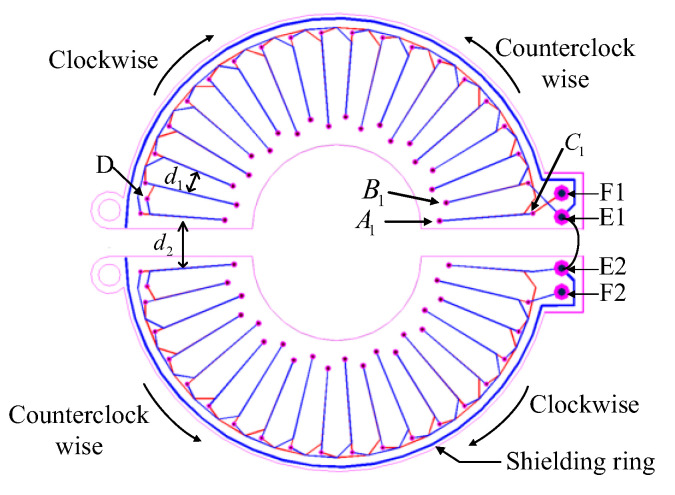
Schematic diagram of the winding mode of the open-ended PCB *Rogowski coil*.

**Figure 14 sensors-22-07979-f014:**
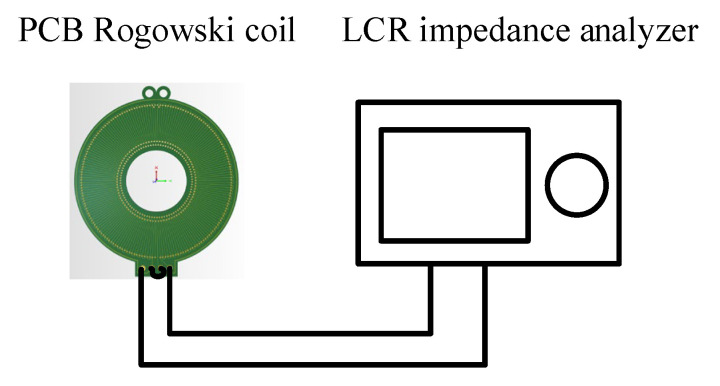
Experimental diagram of the coil electrical parameter measurement.

**Figure 15 sensors-22-07979-f015:**
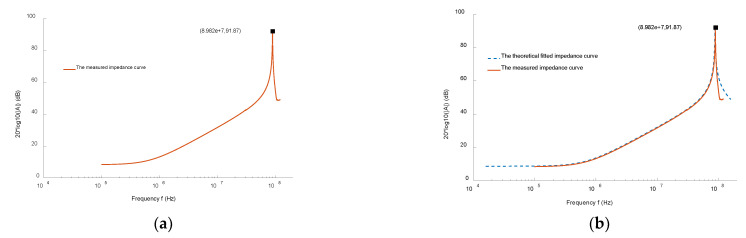
Impedance sweep curve. (**a**) Measured impedance sweep curve, (**b**) theoretical correction and measured anti-sweep curve.

**Figure 16 sensors-22-07979-f016:**
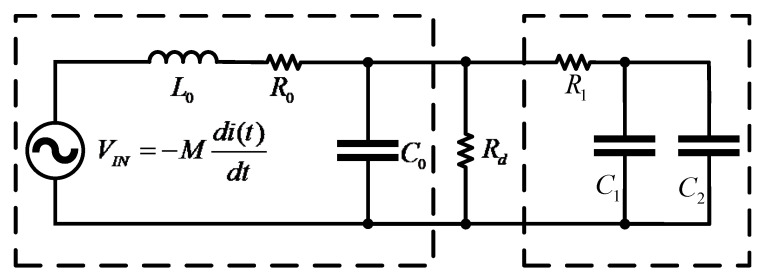
Integrated circuit of the open high-frequency *Rogowski coil* current sensor.

**Figure 17 sensors-22-07979-f017:**
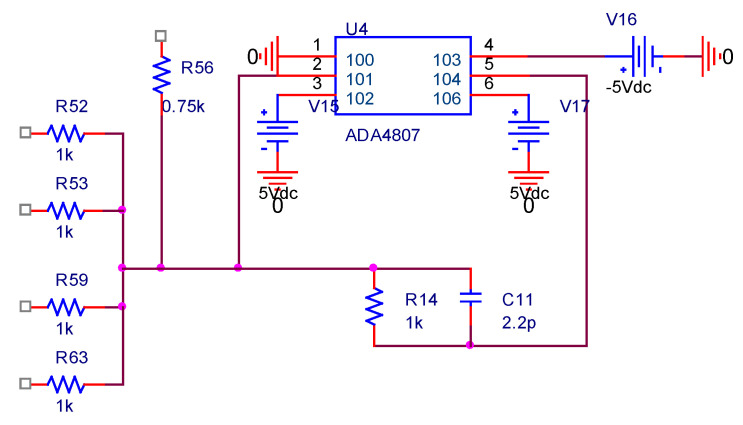
Composite current sensor coupling circuit.

**Figure 18 sensors-22-07979-f018:**
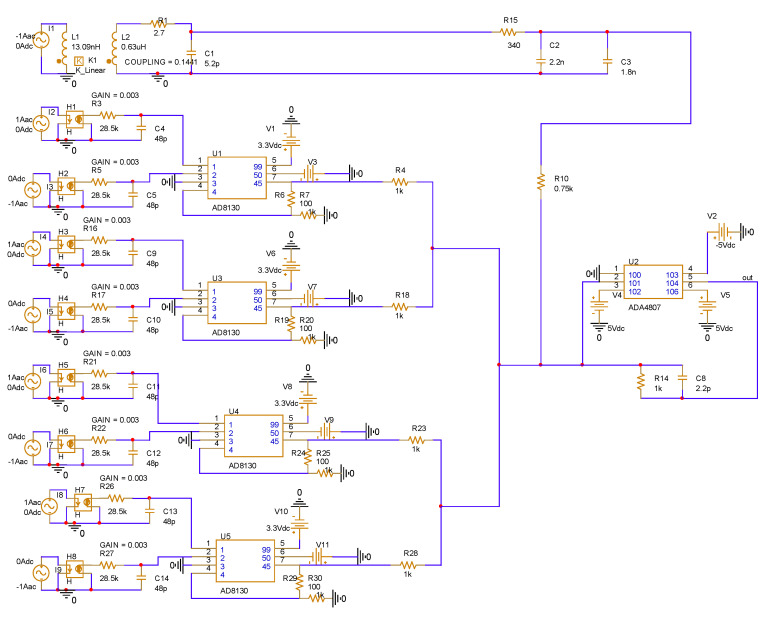
Overall simulation diagram of the contactless composite wide-bandwidth current sensor.

**Figure 19 sensors-22-07979-f019:**
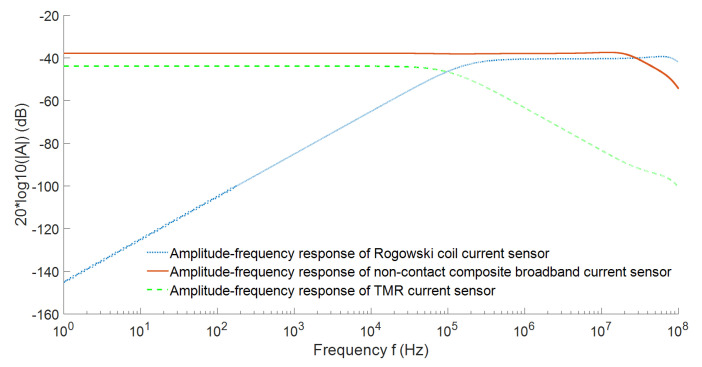
Overall amplitude-frequency characteristics of the non-contact composite wide-bandwidth current sensor.

**Figure 20 sensors-22-07979-f020:**
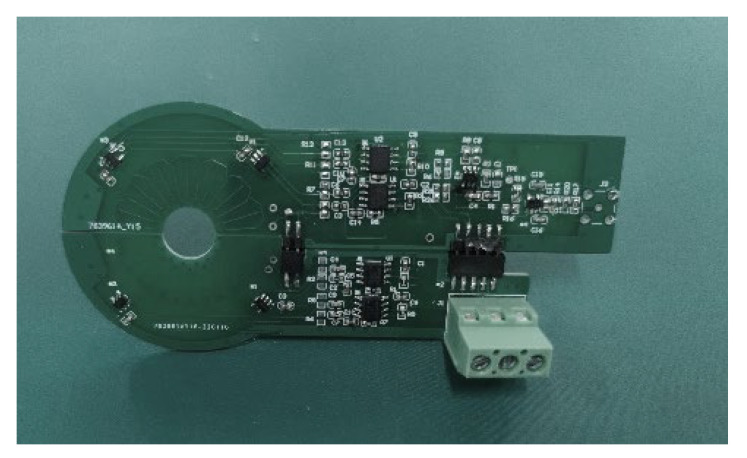
Physical figure of the non-contact composite broadband current sensor.

**Figure 21 sensors-22-07979-f021:**
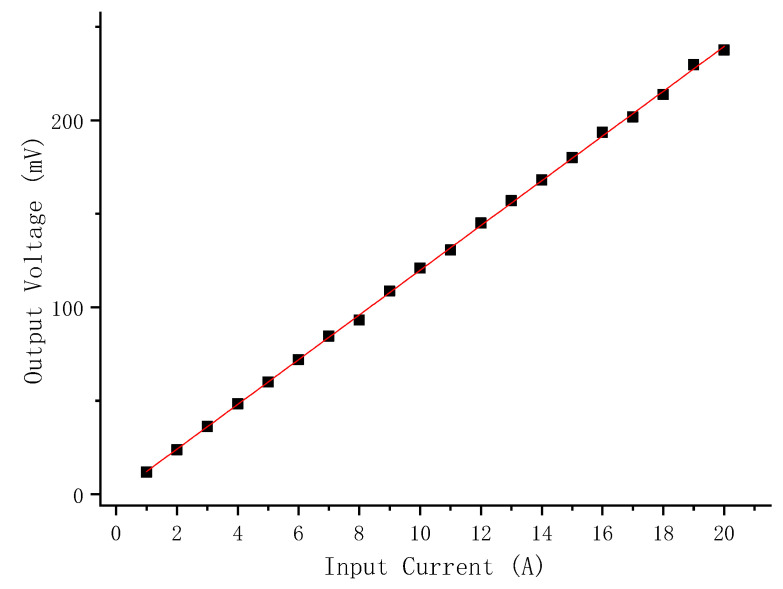
DC input-output curve of the open contactless composite broadband current sensor.

**Figure 22 sensors-22-07979-f022:**
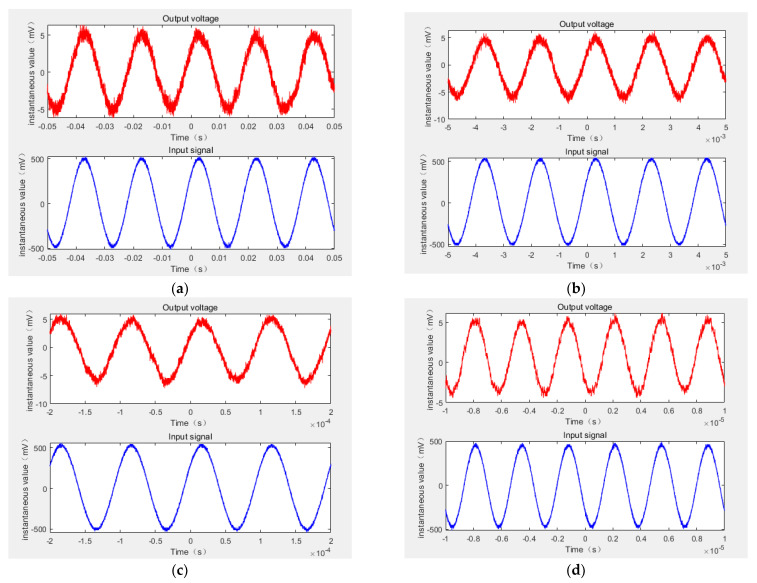
Comparison of the AC input and output of non-contact composite broadband current sensors. (**a**) 50 Hz AC input and output waveform, (**b**) 500 Hz AC input and output waveform, (**c**) 10 kHz AC input and output waveform, (**d**) 300 kHz AC input and output waveform.

**Figure 23 sensors-22-07979-f023:**
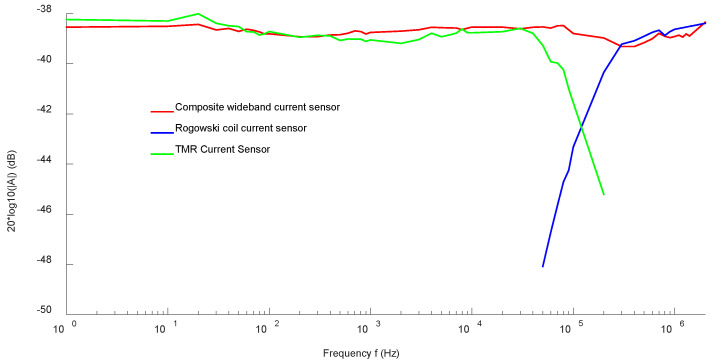
Magnitude-frequency response curve of a non-contact composite broadband current sensor.

**Figure 24 sensors-22-07979-f024:**
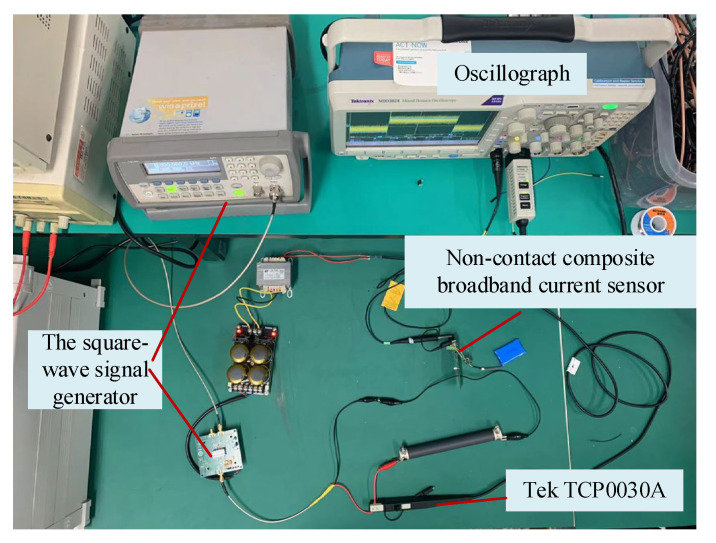
Experimental field diagram of the pulse current testing platform.

**Figure 25 sensors-22-07979-f025:**
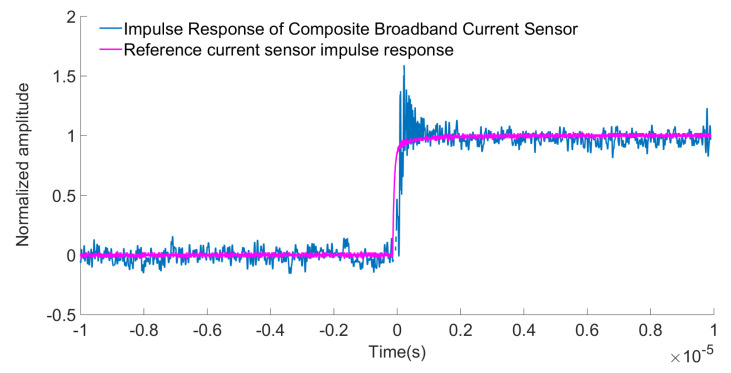
Response diagram of the sensor pulse signal.

**Table 1 sensors-22-07979-t001:** Winding parameters of the coils.

Parameters	Values
Width of winding wire ω	0.1 mm
Thickness of winding wire h	0.035 mm
Number of winding turns N	30
Inner diameter of winding $R_{A}$	7 mm
Inner diameter of winding $R_{B}$	15 mm
Outer diameter of winding b	15 mm
Winding length l	0.4 m
Cross-sectional area of winding A	$3.5\times 10^{-9}m^{2}$

**Table 2 sensors-22-07979-t002:** Lumped parameters of the coil.

Parameters	Values
lumped resistance $R_{0}$	2.7 Ω
lumped inductance $L_{0}$	0.63 μH
lumped stray capacitance $C_{0}$	5.2 pF

**Table 3 sensors-22-07979-t003:** Electrical parameters of composite current sensor coupling circuit.

$R_{52}$	$R_{53}$	$R_{59}$	$R_{63}$	$R_{56}$	$R_{14}$	$C_{11}$
1 kΩ	1 kΩ	1 kΩ	1 kΩ	0.75 kΩ	1 kΩ	2.2 pF

## Data Availability

Not applicable.

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
