# Peer review of "Contactless AC/DC Wide-Bandwidth Current Sensor Based on Composite Measurement Principle"

_sensors, 2022, doi:10.3390/s22207979_

Round 1
Reviewer 1 Report
The article is interesting, has the correct structure and great application value. The literature review is not very extensive, basically only the theoretical background was developed on the basis of the literature. Here, one might be tempted to find similar solutions in world literature. But that's just a suggestion.
I found a few editorial errors described below:
Line 25: "...Rogowski coil has good high-frequency characteristics). and can achieve a wide detection... - The dot before "and" is unnecessary.
Line 38: "...ogy based on capacitive coupling can be used for non-contact measurement of voltage[5-" - space after "voltage".
The entire paragraph (lines 107 through 112) is repeated below Figure 3 (lines 119 through 124).
Lines 196-198: "where a is the sensitivity of the TMR sensor, b is the time constant of the low-pass filter in the low-frequency measurement part, and c is the amplification coefficient of the low-frequency measurement part." - The description concerns the formula (10), but I was wondering for a long time what the parts a, b and c refer to. Formula (10) has no markings of this type.
Line 206 : "Where ..." - "where" should be written in lowercase.
Line 293: "...i.e. d1 = d2 . when the two half-rings..." - the dot before "when " is unnecessary.
Line 354: "Impedance Sweep Curve. (a)Measured Impedance Sweep Curve; (b) Theoretical..." - space after (a).
Line 426: "...ate50Hz, 500Hz, 10kHz, 300kHz sinusoidal current..." - space before 50 Hz.
Line 454: "...broadband current sensor is about-38.5dB." - space after "about".
Reviewer 2 Report
An interesting paper proposes “a non-contact AC / DC composite current sensor based on… tunneling magnetoresistive sensor and… Rogowski coil” in PCB form. The current sensor is a measuring system. Therefore, a comprehensive metrological assessment is essential. The assumption of the 3dB threshold is a too easy condition for measurement applications. There is no description of the test conditions for the proposed current sensor (high-frequency tests are a serious laboratory challenge). There are editorial flaws in the paper, including scheme diagrams.
Selected minor remarks
TMR - “tunneling reluctance”? rather tunnel magneto resistance?
Some of the nodes (in dots form ) are missing from the schematic diagrams, e.g. Fig. 4.
Figs. 8 and 19 - Are the shown responses correctly described?
Units of measurement should be written in normal typeface (instead of italic).
Connections in the scheme diagrams are questionable (e.g. AD8130 connection according to the data sheet).
Fig. 22: Incorrect description of the vertical axes of the graphs, instead of "amplitude" it should be the instantaneous value.
Reviewer 3 Report
Though the paper is well structured and has novel ideas, there are still some problems in this manuscript and it is not suitable for publication now. The comments are as follows:
1. There are some grammatical errors that need to modify such as line 25 in the Abstract section (starting the sentence with and) line 293 (starting the sentence with the small letter “when”) Line 408 (Point position after figure, “Figure 20.”) line 138 (writing where with a capital letter and do not use comma or point after equations in the manuscripts
2. Please clearly itemize the contribution of the paper at the end of the introduction section.
3. Some terms are used and not defined, such as MTJ.
4. For equation (10), there is an explanation in Line 196 but it is not clear where parameters “a, b, and c” are used. There is no explanation about Kh and Kl
5. Suggest including the limitation of this study and future research work.
Reviewer 4 Report
This paper presents one very interesting and practical research.
The Paper is very well organized, with very quality figures.
My comments are as follows:
- First, the Introduction section is short. I want to see the motivation for this paper. Also, please add in Introduction the main paper's contributions.
- Second, the authors use a lot of transfer functions as well as equations. It will be great to present all values of the used variables in Tables, for example in the appendix.
- Third, Fig. 14 is not clear. Please define what it represents.
- Have you used some special instrument for filtering noise in the measured signals?
- Reference list is short. Please cite more literature.
Round 2
Reviewer 2 Report
The paper is improved. Doubts are related to Figs. 8 and 18 - “hybrid sensor amplitude-frequency response” and “Amplitude - frequency Response of Non-contact Composite wideband Current Sensor”. Does the “hybrid sensor” have a higher bandwidth than the separate Rogowski coil and TMR sensors? The description of the metrological properties is still insufficient! Many of the nodes (in dots form) are missing from the schematic diagrams in Figs. 3, 4, 6, 11, 16.
Reviewer 3 Report
The whole comments are almost addressed precisely in the revised version of the manuscript.
